# Adhesive Hemostatic Flake Particulates Composed of Calcium Alginate–Starch–Polyacrylamide/Poly(Acrylic Acid) Ionic Networks

**DOI:** 10.3390/polym17050568

**Published:** 2025-02-20

**Authors:** Yunjeh Ko, Eun Jin Kim, Oh Hyeong Kwon

**Affiliations:** 1Department of Polymer Science and Engineering, Kumoh National Institute of Technology, Gumi 39177, Gyeongbuk, Republic of Korea; yunjehko@gmail.com; 2Theracion Biomedical Co., Ltd., Seongnam 13201, Gyeonggi, Republic of Korea; ejkim@theracion.com

**Keywords:** hemostatic flake, adhesion, calcium alginate, starch, polyacrylamide

## Abstract

Hemostatic particles have specific advantages when applied to narrow and complicated bleeding sites with convenient usage compared to other types of hemostatic agents such as fabrics, foams, and pastes. However, powdery hemostatic agents are easy to desorb from the bleeding surface due to blood flow, which causes a serious decrease in hemostasis function. Here, we introduce bioresorbable flake particulates composed of calcium alginate, starch and polyacrylamide/poly(acrylic acid) ionic networks as a wound adhesive hemostatic agent. The microstructure, chemical characteristics and blood infiltration of the flake hemostatic agent were analyzed. In vitro blood absorption, coagulation ability, adhesion force, cytotoxicity and in vivo bioresorption with biological safety were investigated. The tissue adhesive force of the hemostatic flakes showed a consistently higher value (−0.67 ± 0.06 N axial force) than Arista^TM^ AH powder. The in vivo rat hepatic hemorrhage model analysis demonstrated a significantly improved hemostasis rate in the flake group (36 ± 5 s) by wound adhesion and quick blood absorption. This adhesive flake particulate hemostatic is expected to provide an advanced option for medical treatments.

## 1. Introduction

In recent years, the surgical operations frequency has been rapidly increasing due to the advances in medical devices, technology and infrastructure [1]. Bleeding of local wounds and simple surgical procedures is sufficiently controlled by body hemostasis. However, uncontrolled hemorrhage has serious consequences during precise and complicated surgery. Normal adult blood is about 6~7% of the body weight, and a risk to survival occurs when losing more than 10% of the total blood volume [2]. Therefore, the development of advanced hemostatic agents is essential. When a wound occurs in the body, epithelialization proceeds through hemostasis and an inflammatory phase within the initial day, followed by the proliferation of epithelial cells for 3 weeks [3,4]. 

Hemostatic agents are applied at the earliest stage during medical treatments of wound hemorrhage to prevent the outflow of blood from the body [5]. There are mainly mechanical and chemical methods for hemostasis. Mechanical methods include direct pressure with gauze, sutures, clips, or clamps. Chemical methods include blood coagulants, astringents, and local hemostatic agents. Recently, since blood components have plenty of plasmid, numerous hemostatic agents using quick absorptive polymeric materials have been developed [6,7,8,9].

A drug used as a hemostatic agent absorbs fibrin, fibrinogen, thrombin, and thromboplastin, which are blood and tissue components of the blood clotting system. It reacts with a gelatin sponge or hemoglobin to promote coagulation by accelerating the precipitation of fibrin to coagulate the blood. And oxidative regenerated cellulose (Surgicel^®^, Ethicon, Raritan, NJ, USA), which promotes hemostasis through gauzes or compressed sponges, can be used. But it can cause uncomfortable pressure and pain during packing into bleeding tissues of patients. Therefore, the use of absorbent packing materials in narrow and complicated tissues is not recommended. Representative examples include gelatin sponge (Gelfoam^®^, Pfizer, New York, NY, USA), gelatin/thrombin mixture (Flossal^®^, Baxter, Round Lake, IL, USA), hyaluronic acid sponge (MeroGel^®^/Meropack^®^, Medtronic, Jacksonville, FL, USA), absorbable collagen sponge (Avitene^®^, Davol, Warwick, RI, USA), bioabsorbable polyurethane foam (Nasopore^®^, Polyganics, Groningen, The Netherlands), and carboxymethylcellulose mesh (Sinu-Knit, ArthroCare^®^, Glenfield, UK) [4,5,6]. The commercialized absorptive packing is uncomfortable in abdominal surgery of complex and narrow hemorrhage sites. Therefore, it is necessary to develop a patient-friendly hemostatic agent that has an excellent hemostatic effect in complex and narrow wound hemorrhage sites. A paste or powder type of hemostatic agent (Arista^TM^ AH, BD, Franklin Lakes, NJ, USA) is one of the alternatives. However, the critical problem of powders such as Surgicel-powder (Ethicon, Raritan, NJ, USA), Celox^TM^ Granules (Celox Medical, Crewe, UK), Arista^TM^ AH (BD, Franklin Lakes, NJ, USA), Gelita-Cel-CA powder (Gelita Medical, Baden-Württemberg, Germany), and OOZFIX^TM^ (Theracion Biomedical, Seongnam, Gyeonggi, Republic of Korea) is desorption from the wound site by blood flow [10,11,12,13].

In order to develop an effective hemostatic agent to overcome the above problems, a flake type high absorbent adhesive hemostatic agent containing blood coagulation factors such as calcium chloride was designed using alginate and starch, with biocompatibility and biodegradability, as the main materials. Flake type hemostatic agents are easy to apply even to narrow and complicated areas [13,14,15,16,17,18,19,20]. In addition, it was intended to overcome the problem of a powder hemostatic agent lost during the hemostasis process by providing wound site adhesion. Polyacrylamide is highly water-absorbent, forming a soft gel when hydrated. The major application of polyacrylamide is additives for pulp processing and papermaking. It exhibits high viscosity (stickiness) at a low concentration when dissolved in water.

Since these flake particles have high blood absorption and adhesiveness, they could be applied in various ways and could be widely used in the medical field. In particular, this hemostatic agent has a compressed flake structure for quick blood infiltration, thereby maximizing the effect of blood coagulation.

In this study, an adhesive flake hemostatic agent that has rapid blood absorption and coagulation and wound site adhesion was investigated. This adhesive flake hemostatic agent is expected to have great potential in medical applications.

## 2. Materials and Methods

### 2.1. Fabrication of Alpha-Starch Powder

Rice starch is a plant-originated bioresorbable material [5,11]. The neat rice starch powder was treated with a steam sterilizer to form alpha-starch. In brief, 50 wt% of rice starch (Daejung, Gyeonggi, Republic of Korea) in purified water was gelatinized in a steam sterilizer (at 120 °C, 15 min, MaXterile^TM^ 100, Daihan Scientific, Gangwon, Republic of Korea). The gelatinized alpha-starch paste was spread on a flat plate covered with plastic wrap and placed in a freezer (−80 °C) for an hour. Then, the frozen alpha-starch was lyophilized for 3 days. Then, the dried starch blocks were pulverized using a mechanical grinder (PB2160S, Bomann, Shandong, China). Finally, it was kept in a sealed container before use.

### 2.2. Fabrication of Calcium Alginate–Starch–Polyacrylamide/Poly(Acrylic Acid) Flakes

The flakes with a composition of sodium alginate (Duksan Pure Chemicals, Gyeonggi, Republic of Korea), alpha-starch, polyacrylamide/poly(acrylic acid) copolymer (PAMID, SNF Korea, Seoul, Republic of Korea), and calcium chloride (CaCl_2_, Daejung, Gyeonggi, Republic of Korea) were prepared using the following procedures (Table 1, Figure 1). The starch, alginate and calcium chloride were ground using a mortar. They were blended in water at room temperature with a mechanical mixer (1000 rpm, 5 min, SMX-H230, Shinil Electronics, Seoul, Republic of Korea) for ionic assembly of calcium alginate [13,16]. And then the mixed solution was blended with PAMID using a mechanical stirrer (300 rpm, 3 h, HS-30D, Daihan Scientific, Gangwon, Republic of Korea). After that, the calcium alginate–starch–PAMID hydrogel was lyophilized and pulverized by a mechanical blender (1000 rpm, 1 min, PB2160S, Bomann, Shandong, China). The pulverized powder was compressed using a hydraulic compressor (95 kN, 5 min, SSP-10A, Simadzu, Kyoto, Japan). Then, the particulate sheets were fragmented again and sieved using a series of sieve trays (nominal aperture: 125, 250, 500, 1000, 2000, and 3000 μm, Daihan Scientific, Gangwon, Republic of Korea) to collect homogeneously sorted flake particles. The flake samples were sterilized using UV light and kept in a sealed pack.

### 2.3. Chemical Analysis and Microstructure

The chemical characteristics of the hemostatic flakes were confirmed by using a Fourier transform infrared (FTIR) spectrometer (IRAffiinity-1S, Shimadzu, Kyoto, Japan). All spectra were recorded in an absorption mode with a 4.0 cm^−1^ scan interval in the scanning range 4000~800 cm^−1^.

The morphology of the hemostatic flakes was observed using a scanning electron microscope (SEM, JSM-6380, JEOL Ltd., Tokyo, Japan) with 11~15 kV acceleration voltage range after sputter coating with platinum. The mean diameter (D) of the polygonal flake particles was determined by measuring the equivalent area diameter (D^2^ = 4A/π, where A is the particle area and π is the ratio of the circumference of a circle to its diameter) using image analysis software (ImageJ 1.53e, National Institutes of Health, Bethesda, MD, USA) [13]. Three replicates of each specimen were used for measuring the particle size of the flakes.

### 2.4. Blood Infiltration Test

Dimensional optimization and blood infiltration have critical correlations as particulate-type hemostatic agents. The blood infiltration behavior of the flake particulates was investigated as a function of dimension. The flake particulates (0.1 g) were placed in a glass ring with a silicone sheet. Then, 1 mL whole blood (dog, 13% of anticoagulant, Korea Animal Blood Bank, Gangwon, Republic of Korea) was added. CaCl_2_ (0.0125 M) was dissolved in the blood prior to the experiment to avoid an anticoagulant effect. Thereafter, the blood infiltration was observed after 30 and 120 s.

### 2.5. In Vitro Blood Absorption Amount and Rate

The maximum blood absorption amount of hemostatic flake was measured using the following method. The flake particulates (0.1 g) were placed on a Teflon sheet and whole blood (dog, 13% of anticoagulant, Korea Animal Blood Bank, Gangwon, Republic of Korea) was added until maximum absorption. CaCl_2_ (0.0125 M) was dissolved in the blood prior to the experiment to avoid an anticoagulant effect. Thereafter, the Teflon sheet was flipped to a 45 degree angle to eliminate any non-absorbed blood. The weight of blood absorbed by the flakes was measured and calculated as the absorbed blood.

The blood absorption rate of the hemostatic agent was measured using the following method. The flakes (0.1 g) were placed on a watch glass (37 °C). Then, blood droplets (1000 μL) were added onto the flake particulates. The completion time of blood absorption was recorded.Blood absorption amount (mL/g) = absorbed blood amount (mL)/sample weight (g)

### 2.6. In Vitro Blood Coagulation Time (Lee–White Method)

Quantitative analysis of blood coagulation ability is essential to evaluate the hemostatic function of specimens. To analyze the blood absorption and coagulation characteristics of the flake hemostatic agent, whole blood (dog, 13% of anticoagulant, Korea Animal Blood Bank, Gangwon, Republic of Korea) was used throughout the experiment. CaCl_2_ (0.0125 M) was dissolved in the blood prior to the experiment to avoid an anticoagulant effect.

The blood coagulation time was measured by the Lee–White method [18]. The glass vial was placed in a 37 °C water bath for 10 min to set the surface temperature of the vial. And then, blood (1 mL + 0.0125 M CaCl_2_ 0.1 mL) and hemostatic flakes (0.01 g) were put into the vial and the vial was tilted in every 15 s. The time until there was no longer any flowing blood after contact with the hemostatic agents was recorded.

### 2.7. Tissue Adhesion Ability Test Using Rheometer

The adhesion force of the flake hemostatic agents was measured by using a hybrid rheometer (Discovery HR-2, TA Instruments, Lindon, UT, USA). The specimen was prepared as a gel state by adding PBS (0.5 g) to the particulates (0.1 g) for 1 min of swelling. The porcine skin tissue was fixed onto the upper surface parallel Peltier steel plate (40 mm in diameter, 35 °C) of the rheometer. Then, the specimen was placed on the bottom plate of the rheometer. The upward force was applied after the porcine skin tissue was attached to the top of the rheometer. The adhesion force was recorded through the falling force between the porcine tissue and the gel state specimens.

### 2.8. Cytotoxicity Test

An elution test of bioabsorbable hemostatic particulates was performed using NIH3T3 fibroblasts (mouse embryonic fibroblasts, Korean Cell Line Bank, Seoul, Republic of Korea) to investigate cytotoxicity (ISO 10993-5: 2009, Biological Evaluation of Medical Devices). The specimens were eluted with minimum essential medium (MEM, WELGENE, Gyeongbuk, Republic of Korea) culture solution containing 10% fetal bovine serum (FBS) with 1% penicillin–streptomycin in an incubator (37 °C, 5% CO_2_, 24 h) as extraction medium. Fibroblasts were cultured in low glucose Dulbecco’s modified Eagle’s medium (DMEM, WELGENE, Gyeongbuk, Republic of Korea) supplemented with 10% FBS and 1% penicillin–streptomycin. Cultures were maintained at 37 °C in a humid atmosphere containing 5% CO_2_. When the cells reached 80% confluence, they were detached by 0.25% trypsin containing 1 mM ethylene-diamine-tetraacetic acid (EDTA). And then, cells were seeded with 10,000 cells/well (96 well plate) for 24 h To assess cell viability using the MTT (3-(4,5-dimethylthiazol-2-yl)-2,5-diphenyltetrazolium bromide) assay, the culture medium was removed and replaced with the as-prepared extraction medium and then 100 μL MTT solution was added to each well. After 3 h incubation at 37 °C, the growth and shape of the cells were observed through an optical microscope (ECLIPSE TS100, Nikon Corp., Saitama, Japan). And then, 50 μL dimethyl sulfoxide was added to dissolve the formazan crystals. The dissolved solution was maintained for 15 min and then moved into a 96-well plate. The optical density of the formazan solution was detected by a multi-wall microplate reader (PHOmo, Autobio Labtec Instruments Co., Ltd., Zhengzhou, China) at 450 nm for the cytotoxicity calculation.

### 2.9. In Vivo Hemostasis

Outbred male Sprague–Dawley rats (230~280 g, 8 weeks old, Hyochang Science, Daegu, Republic of Korea) were used as experimental models to evaluate the in vivo hemostasis and bioabsorption. All animal experiments were reviewed and approved by the Institutional Animal Care and Use Ethics Committee of the Konkuk University College of Veterinary Medicine. Experimental procedures were approved by the Animal Care Committee (No. KU21123) and were performed as follows. Rats were anesthetized using an avertin solution (2,2,2-tribromoethanol, Sigma–Aldrich, St. Louis, MO, USA). Following stabilization of the rats, their livers were opened and bleeding was created using a punch biopsy (6 mm). Before applying the hemostatic agents, the initial bleeding was wiped away with PBS and cotton gauze, then the samples (0.1 g) were applied onto the wounded liver. Thereafter, a 50 g weight block was placed onto the site for 10 s. The site was rinsed with saline solution and observed every 15 s.

### 2.10. In Vivo Bioresorbability and Histology

The bioresorbable hemostatic particulates do not require removal as a blood clot complex material. We performed in vivo experiments to evaluate the bioresorbability of the flake particulates. The flake particulates (0.1 g) were inserted into the dermal layer over the rat scapula. After 1, 3, 7, and 15 days from the date of insertion, the animals were necropsied. And samples at the implantation site were visually observed and photographs were taken. For histological evaluation, tissue specimens from the insertion area were collected and fixed for 24 h in 10% formalin solution. The fixed tissue was embedded in paraffin and then sectioned to 5 μm thickness. The tissue sections were deparaffinized and stained with hematoxylin–eosin (H&E) and Masson’s trichrome. The slides were mounted and observed using an optical microscope (ECLIPSE TS100, Nikon, Saitama, Japan) equipped with a digital camera (DS-Fi-2, Nikon, Saitama, Japan). And inflammation was confirmed.

### 2.11. Statistical Analysis

All data are presented in means ± standard deviation. Statistical analysis was performed using data analysis software (KyPlot version 6.0, KyensLab, Inc., Tokyo, Japan). Significance levels were calculated by parametric Student’s *t*-tests and one-way analysis of variance (ANOVA) with a post hoc test based on Tukey’s method. Statistical significance was determined with *p* < 0.05.

## 3. Results and Discussion

### 3.1. Chemical Analysis

The flake hemostatic particulates were prepared by ionic assembly interpenetration gelation with alginate, starch, PAMID and calcium chloride. The functional groups of the hemostatic samples were investigated by FTIR analysis (Figure 2). The base materials of the flake samples are alginate and starch. The spectrum of sodium alginate showed absorption bands from hydroxyl, ether and carboxylic functional groups. Stretching vibrations of O-H bonds of alginate appeared in the range of 3000–3600 cm^−1^. Stretching vibrations of aliphatic C-H were observed at 2920–2850 cm^−1^. Observed bands at 1610 and 1415 cm^−1^ were attributed to asymmetric and symmetric stretching vibrations of carboxylate salt ions, respectively. The characteristic spectra of starch showed broad peaks at 3430 cm^−1^ and 2930 cm^−1^ corresponding to the O-H and C-H_2_ stretching vibrations, respectively. And the peaks at 1660 cm^−1^ were assigned to -OH bending vibration. The peaks at 1000–1200 cm^−1^ originated from -C-O-C- linkage stretching of the polysaccharide backbone chain. The spectrum of PAMID showed absorption bands at 3410–3421 cm^−1^ (NH_2_), 3190–3194 cm^−1^ (NH_2_) and 1682–1685 cm^−1^ (C=O), characteristic of the acrylamide unit. The peak at 1165 cm^−1^ was assigned to C-O bond in poly(acrylic acid). The FTIR spectra of the flake samples showed peaks at 1610, 1660 cm^−1^ and 1415 cm^−1^. And the sharp absorption peak at 3200 cm^−1^ is attributed to O-H stretching. We confirmed that the peak of flake samples is attributed to the -COO^−^ asymmetric and symmetric stretching vibrations with ionic self-assembly of calcium alginate [13,16].

### 3.2. Microstructure of Hemostatic Flakes

Particle surface and structural morphology of a hemostatic flake agent are critical parameters in blood infiltration, absorption and clotting behavior of the base materials. Because the specific surface area of the particles and interfacial voids for quick blood absorption during hemostasis are in proportion to the particle dimension and structural morphology. The lyophilized calcium alginate–starch–PAMID construct was pulverized into a fine powder and compressed under high pressure, then fragments were sieved to obtain homogeneous flakes. The motivation of this study is developing a wound adhesive and effective particulate hemostatic agent. To achieve this goal, we have devised compressed flake particulates that facilitate blood absorption and adhesion. As shown in Figure 3A–F, we prepared six kinds of flake particulates with different dimensions using a multilayered standard sieve. And the photomicrographs of 1 mm flakes (Figure 3G,H) showed flat and compressed flake shape with a relatively rough surface. Singh reported that the irregular, non-spherical particles of a hemostatic powder allowed more void for blood absorption and clot formation [19,20]. We assumed that the flake hemostatic could provide quick absorption of blood plasma and clot formation with a wound adhesive function.

### 3.3. Blood Infiltration on Flake Dimension

One of the critical considerations of particulate type hemostatic agents is dimensional optimization of the particulates for rational hemostasis. Therefore, here we investigated the blood infiltration behavior of the flake particulates as a function of dimension in comparison to Arista^TM^ AH powder. As shown in Figure 4, blood infiltration was facilitated with increasing particulate dimension. The flake dimensions of 0.5 mm and 0.25 mm were not infiltrated with blood. And the Arista^TM^ AH was not infiltrated with blood even after 120 s. Meanwhile, the flake dimensions of 1 mm, 2 mm, and 3 mm facilitated blood infiltration under 30 s. The gap space of the accumulated particulate is increased in proportion to the particulate dimension. Therefore, we assumed that the flake dimension of 1 mm was best for further tests.

### 3.4. In Vitro Blood Absorption and Coagulation Time

The in vitro evaluation is a method that measures the blood absorption amount over time until maximum absorption by contacting hemostatic agents. As shown in Figure 5, the blood absorption amount and absorption rate of both A5S5-PC flake and Arista^TM^ AH powder were similar and less than 5 s. Blood absorption amount and rate are in proportion to the source materials and the specific surface area [3,4,5]. Therefore, we assumed that the blood absorption amount and rate were significantly improved by the combination of calcium alginate and starch with PAMID.

The short coagulation time means the effective activation of blood coagulation factors and the greater adhesion of platelets. As shown in Figure 6, the blood coagulation time of the A5S5PC group showed a significant decrease compared to the Arista^TM^ AH group. Even if the blood absorption amount of the hemostatic powders was similar, the A5S5-PC group showed quick blood coagulation compared with the other groups. The A5S5-PC maintains a network structure after blood absorption by ionic assembly with calcium chloride, promoting blood coagulation. We assumed that the calcium ions on the alginate, starch, and PAMID network structure facilitated the hydrophilicity and blood coagulation with gelation during the latter steps of hemostasis.

### 3.5. Tissue Adhesion Ability

The powder hemostatic agents have been widely adopted for their convenience, such as in narrow and complicated sites, ease of use, and bioresorbable agents. However, their efficacy has limitations due to their partial loss at the bleeding site compared to other types (fabric, sponge, gel) of hemostatic agents [5,11]. A tissue adhesive hemostatic particulate could allow fair hemostatic ability without a serious loss of hemostatic agents. As shown in Figure 7, it was confirmed that the maximum adhesive strength of the A5S5-PC group (−0.67 N) was superior compared to the other groups. The A1S9-PC group (−0.30 N) showed the lowest adhesion value. The A7S3-PC and A1S9-PC groups showed −0.45 N and −0.40 N, respectively. The alginate leads to an initial rapid absorption, and starch leads to a latter gradual absorption. And the adhesion force of the A5S5-PC group was twice as strong as that of the Arista^TM^ AH group (−0.25 N) with a long duration. The A5S5-PC flake is composed of calcium alginate, starch, and PAMID. The calcium alginate and starch increase the hydrophilicity with gelation by absorption of blood plasma. In particular, the PAMID has a very sticky property after water absorption. It will facilitate the adhesion force of A5S5-PC flake samples.

### 3.6. Cytotoxicity

Quantitative cytotoxicity of flake hemostatic agents was evaluated by following the Biological Evaluation of Medical Devices (Figure 8). Compared with Arista^TM^ AH (87%), the cell viability of all flake groups was 94~81%. This means that the flake hemostatic agent is allowed for medical applications. The qualitative analysis was carried out through microscopic observation of a cell monolayer. The flake groups were barely observed to separate from the extracellular matrix and inhibit cell growth, and showed grade 2. Although foreign body reactions to natural polymers are rare, investigating the cytotoxicity of a flake particulate product made of calcium alginate, starch, and PAMID is important. In both quantitative and qualitative evaluation, the flake hemostatic agent was safe to apply for medical uses.

### 3.7. In Vivo Bioresorption and Histology

The hemostatic flake particulate was designed as a bioresorbable agent after completing hemostasis without the need for removal. Thus, the bioresorption behavior of the particulate was investigated in an in vivo environment (intradermal insertion over the scapula of rats) through observation of the morphology and histology after a set period. The flakes remained after 7 days with partial debris, while most of the flakes were resorbed within 15 days (Figure 9). Otherwise, the Arista^TM^ AH group showed quicker bioresorption behavior than the flake group because the base material of Arista^TM^ AH is carboxymethyl starch which is a water-soluble, biodegradable and non-toxic polysaccharide. And the Arista^TM^ AH powder has a high specific surface area that facilitates bioresorption. Serious inflammation was not observed in the interface of the dermis and subcutaneous fat layers on histological investigation (Figure 10). However, a large amount of flake particulates was observed remaining until day 7. On the other hand, in the Arista^TM^ AH group, the samples were almost completely resorbed after day 3, and inflammation was not observed.

The bioresorption rate of hemostatic agents varies depending on the source materials (gelatin, chitosan, hyaluronic acid, alginate, starch, oxidized cellulose) and type (fabric, foam, paste, powder). The flake particulates are composed of calcium alginate and starch, which are plant-based bioresorbable materials. The significant hydrolysis and biodegradation were retarded by calcium alginate ionic networks and compressed flake particulates. However, the flake hemostatic agent induced gradual bioresorption within 2 weeks. Therefore, it could relieve any uncomfortable feeling without a foreign body reaction after application during laparoscopic surgery in narrow and complicated areas after completing wound regeneration.

### 3.8. In Vivo Hemostasis Evaluation in Rat Hepatic Hemorrhage Model

Hemostatic evaluation through in vivo animal model experiments is the most important factor in analyzing hemostatic agent performance. As shown in Figure 11A, the average hemostasis time of each group treated with the gauze, A5S5-PC flake, and Arista^TM^ AH powders was 195 ± 54, 36 ± 5, and 104 ± 18 s, respectively. The A5S5-PC flake group showed a crucial reduction in hemostasis time. The rat hepatic hemorrhage site was clear compared to the gauze group (Figure 11B) because the wounded site was rapidly clogged by flake particulates without serious hemorrhage. Various commercialized powder hemostatic agents contain both blood absorption materials (alginate, starch, collagen, gelatin, hyaluronic acid, carboxymethyl cellulose, oxidative regenerated cellulose) and blood clotting reagents (thrombin, CaCl_2_) to facilitate coagulation [4,5,10,11]. The A5S5-PC flake is composed of an ionic assembly of alginate with CaCl_2_ with a starch chain and PAMID as materials. The excellent hemostatic ability of the A5S5-PC flake was induced by the synergetic effect of alginate (initial plasmid absorption), starch (latter plasmid absorption), PAMID (wound tissue adhesion) and calcium (ionic assembly and chemical coagulation). In addition, the A5S5-PC has a compressed particulate flake structure. The flake morphology possess two functions such as (I) quick blood infiltration by weight and void space, and (II) prevention of blood desorption by flake morphology. For these reasons, we have assumed that the calcium alginate–starch–PAMID compressed flake could be a novel adhesive hemostatic agent.

## 4. Conclusions

Flake hemostatic particulate agents composed of calcium alginate, starch, and polyacrylamide/poly(acrylic acid) were fabricated by ionic assembly gelation, lyophilization, pulverization, and compression procedures. The novel microstructure with an optimized formulation of the compressed flakes (A5S5-PC sample, sieve aperture: 1 mm, mean diameter 1.31 mm) facilitates blood infiltration. The amount and rate of blood absorption of A5S5-PC flakes was superior to the other formulation samples and Arista^TM^ AH powder. The in vitro blood coagulation time of the flakes was significantly faster than that of Arista^TM^ AH. In particular, the adhesive strength of the A5S5-PC sample was −0.67 N, which is approximately double the value of Arista^TM^ AH. The cytotoxicity of the flake hemostatic was reasonable. The flake hemostatic was bioabsorbed within 2 weeks after rat scapula intradermal insertion without a serious inflammatory reaction. An in vivo rat hepatic hemorrhage model demonstrated the critical hemostatic ability of the flake hemostatic. This tissue adhesive hemostatic flake will provide a beneficial option for hemostasis treatment.

## Figures and Tables

**Figure 1 polymers-17-00568-f001:**
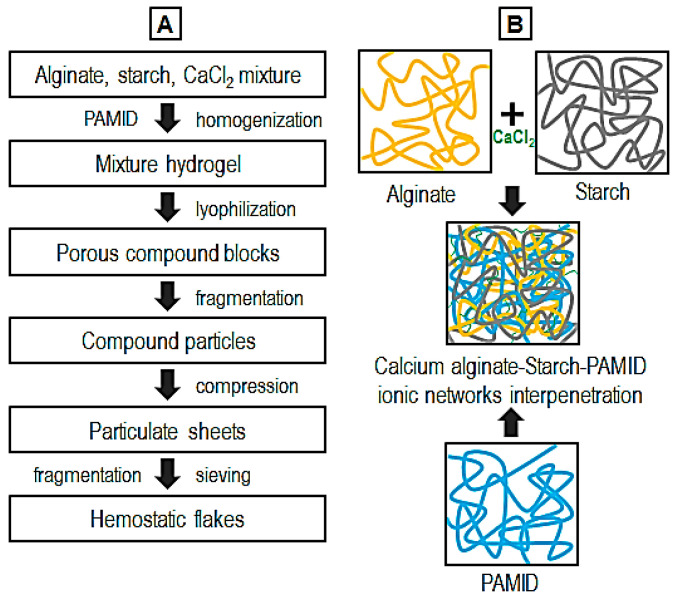
Fabrication procedures (**A**) and gelation schematic (**B**) of the hemostatic flake particulate samples.

**Figure 2 polymers-17-00568-f002:**
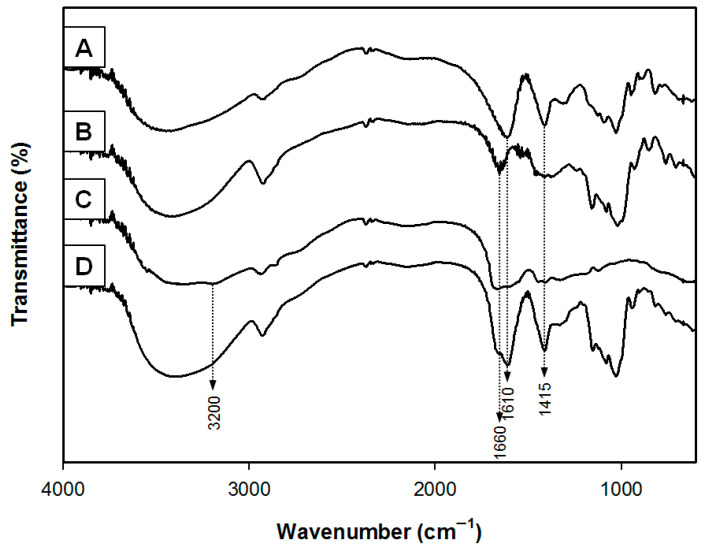
FTIR spectra of (**A**) alginate, (**B**) starch, (**C**) PAMID and (**D**) calcium alginate–starch–PAMID compound.

**Figure 3 polymers-17-00568-f003:**
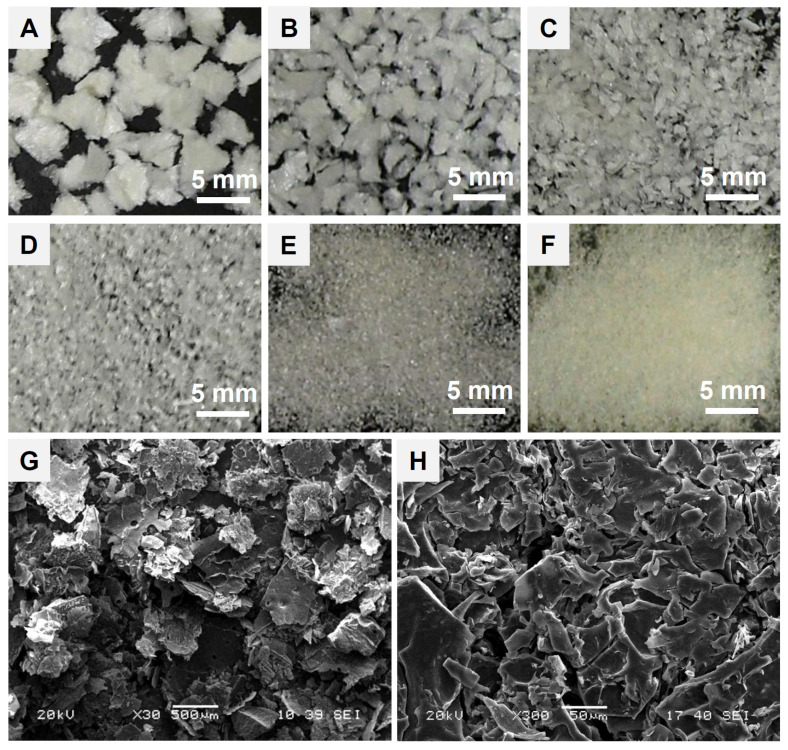
Photographs of the hemostatic flakes (calcium alginate–starch–PAMID compounds) as a function of the sieve apertures (nominal aperture of standard test sieve: ASTM E11) layers of (**A**) 3 mm, (**B**) 2 mm, (**C**) 1 mm, (**D**) 0.5 mm, (**E**) 0.25 mm, and (**F**) 0.125 mm. Photomicrographs of magnified hemostatic flakes in the 1 mm sieve apertures layer ((**G**) ×30 and (**H**) ×300 magnification).

**Figure 4 polymers-17-00568-f004:**
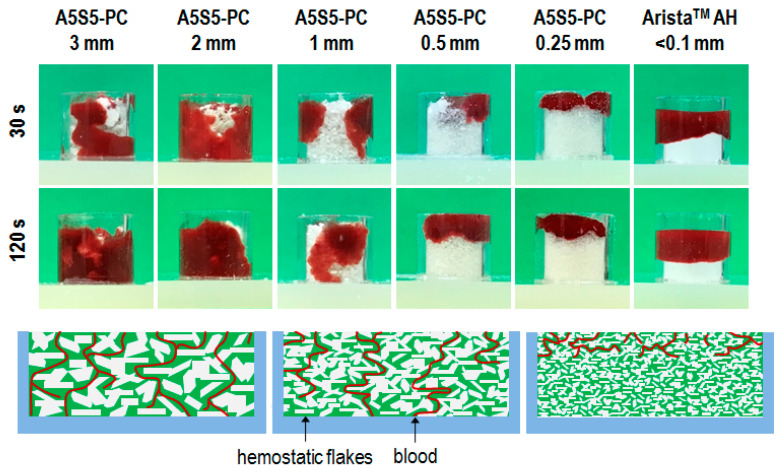
In vitro blood infiltration behavior of the hemostatic flakes (0.1 g) of different particulate sizes and the Arista^TM^ AH (0.1 g) hemostatic agent after 30 and 120 s of blood dropping (1 mL) (n = 3).

**Figure 5 polymers-17-00568-f005:**
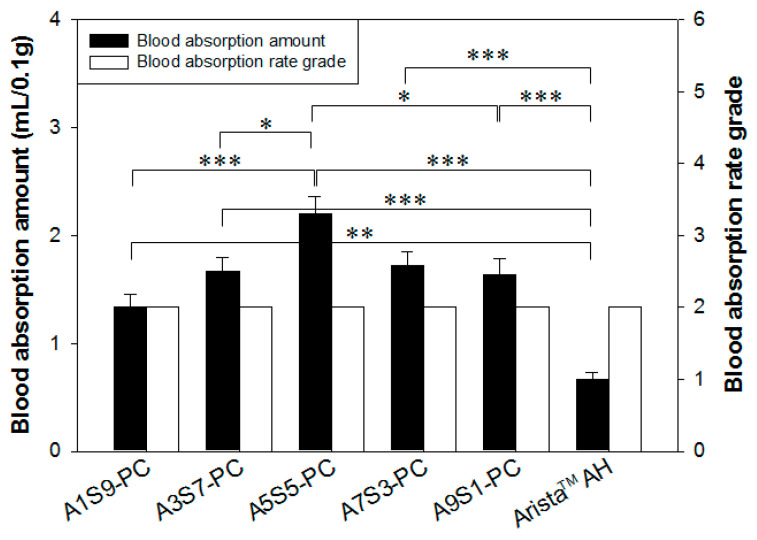
In vitro blood absorption amount and blood absorption rate grade of the hemostatic flakes (sieve aperture: 1 mm) by the ratio of alginate to starch compared with Arista^TM^ AH (n = 3, * *p* < 0.05, ** *p* < 0.01, *** *p* < 0.001). Grade 1: <5 s, Grade 2: 5~10 s, Grade 3: 10~30 s, Grade 4: >40 s.

**Figure 6 polymers-17-00568-f006:**
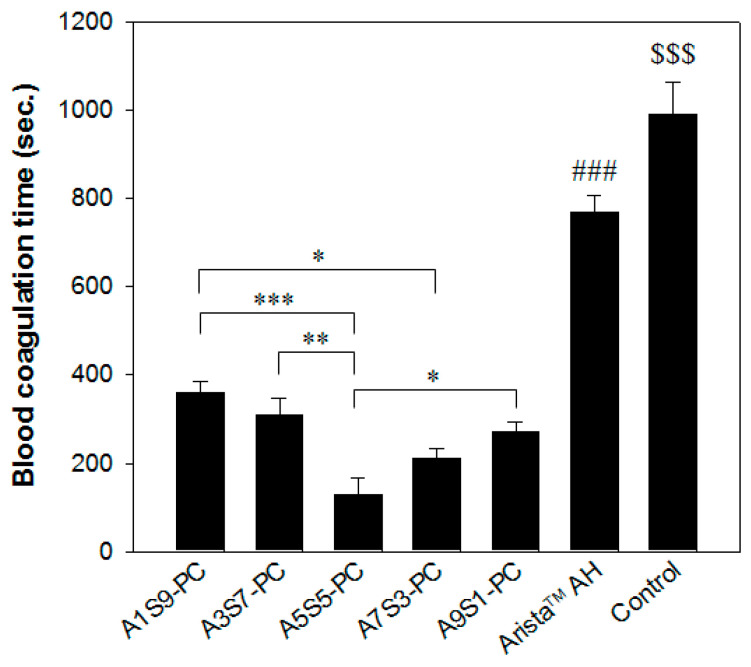
In vitro blood coagulation time of the hemostatic flakes (sieve aperture: 1 mm) on the ratio of alginate to starch compared with Arista^TM^ AH (n = 3, Lee–White method) (* *p* < 0.05, ** *p* < 0.01, *** *p* < 0.001, ### *p* and $$$ *p* < 0.001 with all flake groups).

**Figure 7 polymers-17-00568-f007:**
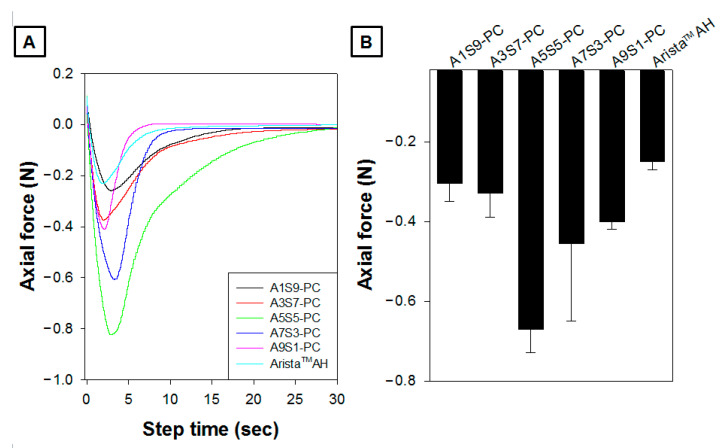
Adhesion force graph as a function of step time (**A**) and maximum adhesion axial force value (**B**) of hemostatic flakes (sieve aperture: 1 mm) by the ratio of alginate to starch compared with Arista^TM^ AH after PBS absorption on porcine skin (gel state of specimens containing 7.5 wt% PBS, n = 3).

**Figure 8 polymers-17-00568-f008:**
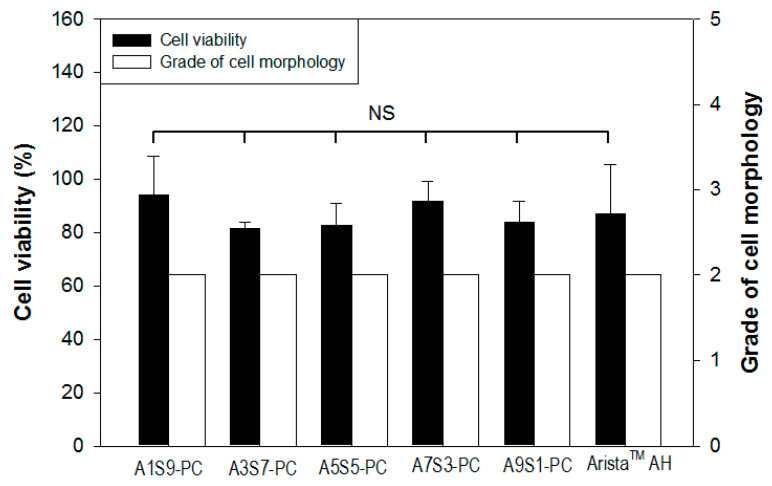
Cytotoxicity evaluation of the hemostatic flakes (sieve aperture: 1 mm) by the ratio of alginate to starch compared with Arista^TM^ AH (n = 3, NS means not significant).

**Figure 9 polymers-17-00568-f009:**
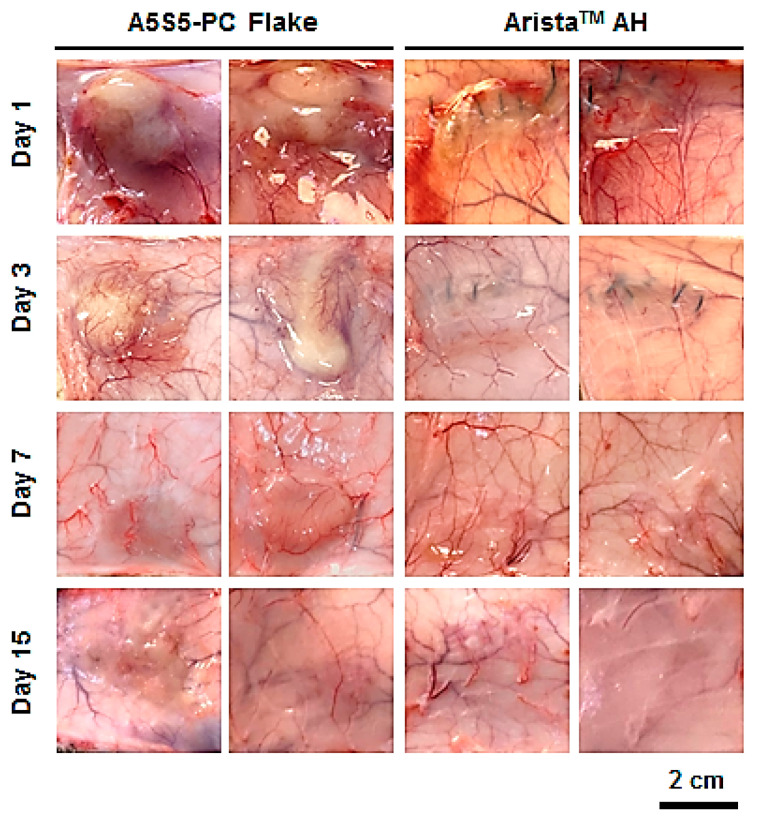
Photographs of in vivo bioresorption of the hemostatic flakes (A5S5-PC, sieve aperture: 1 mm) and Arista^TM^ AH after intradermal insertion over the scapula of rats for 1, 3, 7, and 15 days (0.1 g, n = 4).

**Figure 10 polymers-17-00568-f010:**
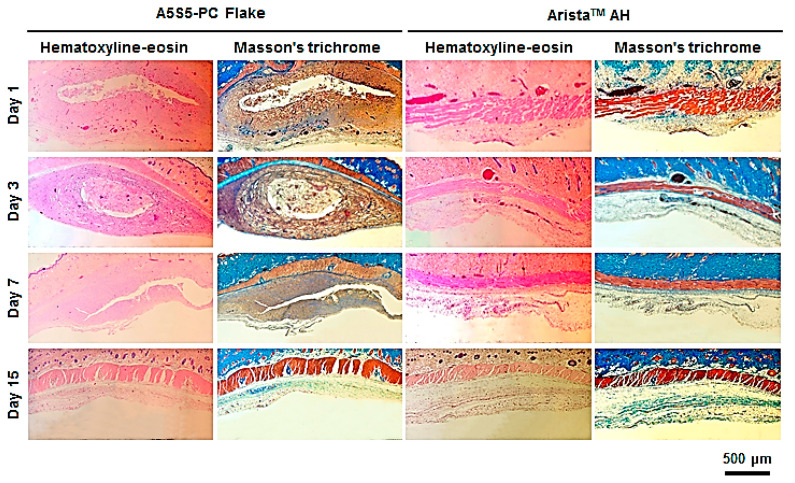
Histology of in vivo absorption of the hemostatic flakes (A5S5-PC, sieve aperture: 1 mm) and Arista^TM^ AH after intradermal insertion over the scapula of rats (0.1 g, n = 4) for 1, 3, 7, and 15 days. Photomicrographs of stained slides of the implanted tissue sites. H&E (pink cytoplasm, dark blue nuclei), Masson’s trichrome (dark red keratin and muscle fibers, pink cytoplasm, light blue collagen, dark blue nuclei).

**Figure 11 polymers-17-00568-f011:**
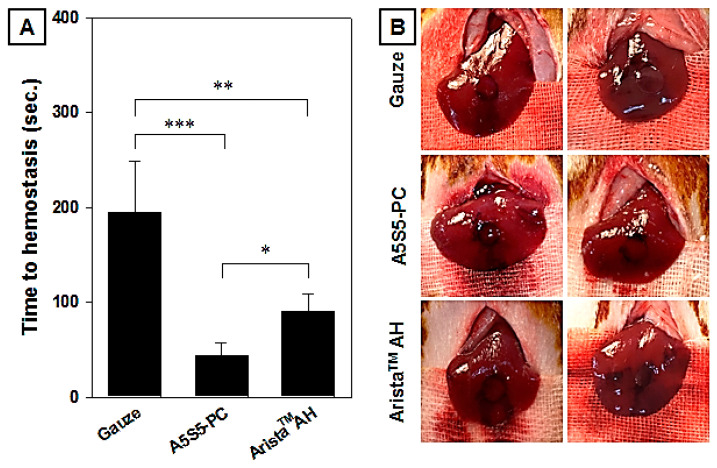
In vivo hemostasis time (**A**) and photographs (**B**) of hemostatic agents applied in the rat hepatic hemorrhage model (0.2 g, n = 5, *** *p* < 0.001, ** *p* < 0.01, * *p* < 0.05).

**Table 1 polymers-17-00568-t001:** Preparation formulations of the hemostatic flakes composed of alginate, starch, polyacrylamide/poly(acrylic acid) copolymer (PAMID), and calcium chloride (CaCl_2_).

Samples	Alginate (wt%)	Starch (wt%)	PAMID (wt%)	CaCl_2_ in Alginate (wt%)	Water (wt%)
A1S9-PC	0.5	4.5	0.5	0.05	94.5
A3S7-PC	1.5	3.5	0.5	0.05	94.5
A5S5-PC	2.5	2.5	0.5	0.05	94.5
A7S3-PC	3.5	1.5	0.5	0.05	94.5
A9S1-PC	4.5	0.5	0.5	0.05	94.5

## Data Availability

Data are contained within the article.

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
