# Peer review of "Adhesive Hemostatic Flake Particulates Composed of Calcium Alginate–Starch–Polyacrylamide/Poly(Acrylic Acid) Ionic Networks"

_polymers, 2025, doi:10.3390/polym17050568_

Round 1

Reviewer 1 Report

Comments and Suggestions for Authors

The study described the the manuscript is elaborate following major corrections will enhance the quality of the manuscript

1. Major references are in the Introduction, please support the methodology and result discussion part with relevant references.

2. While designing the powder formulation please explain the significance of polyacrylamide. What will be its fate after biosorption?

3. Methodology - section 2.8 line 168 authors mentioned 'as-prepared extraction medium' but have not explained how it was prepared?

4. section 2.9 line 184-185 authors applied cotton gauze on the wound before flake formulation, why? what will happen to gauze during flake biosorption.

5. section 2.10 - line 192 what is intradermal scapula of rat? why flakes are inserted intradermally where blood supple is limited? Line 194 - not clear 'evaluation of 2 sites for each breeding day'? New sentence starting with 'And'

6. Reference 12 missing

7. Results Section 3.6 - line 312, 313 'elute of the test substance' please explain it properly either in methodology or in this section. Line 317 - was evaluated by .....' this line is not clear, do you want to mention any standard here?

8. Section 3.7 - line 338, 339 'There was .....' the sentence is not clear.

9. section 3.8 line 374 and 375 - 'alginate (initial .....), starch (latter .....) sentence not clear. line 367 sentence starting with 'And'.

10. Conclusion need to be improved, many sentences are starting with 'And'.

Comments on the Quality of English Language

The English language need to be significantly polished, there are many sentences starting with 'And' and many sentences are without meaning in the text.

Reviewer 2 Report

Comments and Suggestions for Authors

In this article entitled (Adhesive hemostatic flake particulates composed of calcium alginate-starch-polyacrylamide/poly(acrylic acid), the manuscript is interesting however, some aspects should be better explored and explained.

Comments

·        The abstract is very concise, please improve it.

·        The manuscript needs citation of the previous studies. Number of reference very little

·        The result and discussion part lake the interpretation of the results and comparison with the previous findings.

·        Did the authors study the safety of the compound of flaks when they come in contact with the blood? What about the long-term safety?

·        Why authors didn’t make a control group and only compare A5S5-PC sample with AristaTM AH powders?

·        Could you elaborate on how the microstructure of the A5S5-PC flakes facilitates blood infiltration?

·        How did authors ensure that the hemostatic effect was due to the prepared flakes not due to other factors? You should make a control group without a hemostatic agent.

·        In vivo study should include more than one common hemostatic agent other than A5S5-PC flakes to provide a more comprehensive comparison.

Round 2

Reviewer 1 Report

Comments and Suggestions for Authors

NA

Comments on the Quality of English Language

NA

Reviewer 2 Report

Comments and Suggestions for Authors

Authors have  addressed all my comments and the manuscript now is suitable for publication